# “Diving into the Gray Zone”: A Case Report of a 19-Year-Old Patient Treated with Tooth-Borne Rapid Maxillary Expansion

**DOI:** 10.3390/healthcare13222854

**Published:** 2025-11-10

**Authors:** Valentina Coviello, Davide Gentile, Edoardo Staderini, Andrea Camodeca, Angela Guarino, Massimo Cordaro

**Affiliations:** 1Postgraduate School of Orthodontics, Catholic University of the Sacred Heart, Largo Agostino Gemelli 8, 00168 Rome, Italyedoardo.staderini@unicatt.it (E.S.);; 2Department of Chemical Science and Technologies, Materials for Sustainable Development—Dentistry, University of Rome Tor Vergata, 00133 Rome, Italy; 3UOC Clinica Odontoiatrica, Dipartimento di Neuroscienze, Organi di Senso e Torace, Fondazione Policlinico Universitario A. Gemelli—IRCSS, Largo Francesco Vito 1, 00168 Roma, Italy

**Keywords:** adult orthodontics, rapid palatal expander, bilateral crossbite, breathing disorders

## Abstract

**Background:** This case report aimed to quantify dental, alveolar, and skeletal changes, periodontal health, and sleep quality after treatment with a tooth-borne rapid palatal expander (RPE) in a young adult with bilateral posterior crossbite due to transverse maxillary deficiency. Tooth-borne RPE is typically indicated during the prepubertal or pubertal growth phases; however, some post-pubertal or young adult patients may still present with incomplete maturation of the midpalatal suture—the so-called “gray zone.” In clinical practice, treatment decisions should ideally consider multiple skeletal resistance areas (the zygomaticomaxillary buttress, the pterygomaxillary junction, the nasal aperture pillars), although midpalatal suture assessment often remains central to case selection. **Methods:** A 19-year-old male patient presented with a skeletal Class III tendency, dental crowding, and anterior and bilateral posterior crossbites, accompanied by snoring and breathing difficulties. The patient declined surgical- and miniscrew-assisted RPE. Cone-beam computed tomography (CBCT) scan revealed incomplete midpalatal suture maturation. Based on periodontal evaluation, a conventional tooth-borne RPE was chosen. Pre- and post-expansion CBCT scans were used to evaluate dental, skeletal, and periodontal outcomes. **Results:** After one year of treatment, bilateral posterior crossbite was successfully corrected. Buccal bone thickness showed a slight reduction only on the upper left first molar (from 1.2 mm to 0.9 mm), without evidence of dehiscence or fenestration. A 2° increase in the dental tipping angle (DTA) was observed on both molars, and the palatal alveolar angle (PAA) increased by 3°. Sutural separation expanded from 0.32 mm to 7.82 mm. The Midpalatal Opening Related to Expander Opening (MORE) factor was 0.54, indicating a predominantly skeletal response. Periodontal health remained stable, and CBCT analysis confirmed increases in intermolar width (from 36.08 mm to 50.02 mm) and palatal maxillary width (from 28.04 mm to 34.5 mm). A reduction in the Pittsburgh Sleep Quality Index (PSQI) from 7 to 3 was observed, though this finding should be interpreted cautiously due to its subjective nature and the absence of objective airway measurements. **Conclusions:** The present case report suggests that tooth-borne RPE may represent a viable and minimally invasive option for correcting posterior crossbite in carefully selected young adults with incomplete midpalatal suture maturation. However, the findings are limited to a single case with short follow-up and should be regarded as hypothesis-generating rather than conclusive.

## 1. Introduction

Maxillary transverse deficiency is characterized by an inadequate transverse relationship between the maxillary and mandibular teeth, and it affects approximately 10% of individuals in the permanent dentition stage [1]. Since spontaneous correction is not expected, early intervention using an RPE is indicated, particularly during the prepubertal phase [2].

A recent Delphi study proposed that bone-anchored expanders are effective in producing maxillary skeletal expansion in late adolescent patients, whose chronological age thresholds are typically 14–15 years for females and 15–16 years for males. In young adults (until 24 years of age), bone-borne RPE could reduce the risk of buccal inclination of expanded posterior teeth compared to tooth-borne RPE [3]. In clinical practice, treatment decisions should ideally consider multiple skeletal resistance areas (the zygomaticomaxillary buttress, the pterygomaxillary junction, the nasal aperture pillars), although midpalatal suture assessment often remains central to case selection. Advances in clinical and radiographic research have contributed to a more precise understanding of skeletal resistance and treatment predictability in late adolescent and young adult patients. Indeed, a CBCT-based evaluation of midpalatal suture morphology has been shown to provide more reliable information than chronological age when determining the feasibility of non-surgical expansion. Angelieri et al. demonstrated that cervical vertebral maturation (CVM) stages showed a good correlation with midpalatal suture morphology in the prepubertal period. However, in postpubertal individuals, a so-called “gray zone” exists, in which clinical decision-making is more complex: specifically, at CVM stage CS4, there is a moderate probability of encountering suture stage C, while stages D and E are less predictable; indeed, the authors highlighted the importance of a radiographic assessment of the midpalatal suture maturation in postpubertal patients to support the choice between conventional tooth-borne RPE and surgical alternatives such as surgically assisted rapid palatal expansion (SARPE) or miniscrew-assisted rapid palatal expansion (MARPE) [4]. These observations support the concept that, even in post-pubertal individuals with incomplete sutural fusion, a conventional tooth-borne expander may achieve clinically meaningful skeletal effects if appropriate diagnostic criteria and activation protocols are applied. The present case report provides a pragmatic diagnostic framework integrating CBCT-guided evaluation of midpalatal suture maturation and periodontal status to guide case selection in the post-pubertal “gray zone” for non-surgical RPE.

The present study reports a case of tooth-borne RPE in a young adult with transverse maxillary deficiency and bilateral posterior crossbite. Drawing on recent clinical evidence, this case explores the feasibility of this non-surgical approach. The primary objective was to evaluate periodontal health following RPE, while secondary aims included quantifying dental, alveolar, and skeletal changes, as well as assessing the impact of RPE on sleep quality.

## 2. Case Presentation, Diagnostics and Intervention

### 2.1. Patient Information and History

This case report was approved by the Institutional Review Board (0000550/25 17/04/25, Supplementary Material S1) and conducted in accordance with the Declaration of Helsinki (1975) and its subsequent amendments. Written informed consent was obtained from the patient. The report complies with the CARE guidelines (Supplementary Material S2). This patient was selected based on (i) incomplete midpalatal suture maturation (Stage C), (ii) stable periodontal health, and (iii) refusal of surgical approaches. These criteria may critically affect the success of tooth-borne RPE in young adults, as previously suggested by Angelieri et al. [4] and Brunetto et al. [5].

### 2.2. Diagnostic Assessment and Etiology

A 19-year-old Caucasian male was referred to the Foundation “Agostino Gemelli” Teaching Hospital IRCCS, complaining of nocturnal snoring, nasal breathing difficulties during exercise, and a sensation of limited tongue space in the maxillary arch. The patient was in good general health (ASA I), with no familial history of bilateral posterior crossbite or prior orthodontic treatment during childhood or adolescence. A CBCT scan (voxel size 0.3 mm, field of view 16 × 13 cm) was acquired before treatment. Standard pre-treatment lateral cephalogram and orthodontic clinical pictures were obtained, and periodontal charting was performed.

### 2.3. Clinical Findings and Imaging

Extraoral frontal evaluation revealed mandibular symmetry, reduced bizygomatic width, pronounced buccal corridors during smiling, and competent lips at rest. The patient displayed a mild gummy smile, medium smile line, and convex smile arc. Lateral profile was straight, with symmetrical facial heights, normal chin-throat length, and retruded upper incisor display on smiling.

Intraorally, a complete permanent dentition was present, with bilateral Class I molar and canine relationships. Bilateral posterior crossbite affected teeth 1.6, 1.7, 2.4, 2.5, 2.6, and 2.7, while anterior crossbite was observed at teeth 1.2 and 2.2, presenting a negative overjet of 1 mm. The maxillary arch was narrow and V-shaped, contrasting with mild anterior crowding in the mandibular arch (Figure 1).

Both the Spee and Wilson curves were deep; overjet was normal, and overbite reduced. Panoramic radiography showed no dental anomalies.

### 2.4. Cephalometry

Cephalometric analysis (Figure 2) confirmed a skeletal class I, with a skeletal tendency towards Class III (SNA 80.0°, SNB 80.2°, and ANB −0.1°). Although the ANB angle was close to 0°, suggesting a nearly straight skeletal profile, both the Wits appraisal (−0.1) and selected McNamara parameters (A to N–Perp and Pog to N–Perp) indicated reduced maxillary retrusion and mild mandibular prominence. The Wits appraisal (−0.1) supported this interpretation, while FMA (20.1°) indicated a hypodivergent facial growth pattern. The interincisal angle (120.4°) suggested a flaring of the lower incisors. These data are summarized in Table 1.

No mandibular lateral deviation or temporomandibular joint dysfunction signs were noted.

Sleep quality was assessed using the PSQI, which evaluates seven components: subjective sleep quality, sleep latency, duration, efficiency, disturbances, use of sleep medication, and daytime dysfunction. Each component is scored 0–3, yielding a total score up to 21, where higher values indicate poorer sleep quality. A score above 5 distinguishes poor sleepers [6]. The patient scored 7, indicating impaired sleep quality.

### 2.5. Periodontal Status

The patient had no active periodontal disease as shown by the pre-treatment periodontal records (Table 2): modified gingival index, calculus index, plaque index, clinical attachment level (CAL) and bleeding on probing (BoP) [7]. After the initial periodontal evaluation at T0, the patient was instructed and motivated to standard oral hygiene and underwent professional scaling before the orthopedic therapy.

### 2.6. Evaluation of the Skeletal Maturation

According to cervical vertebral maturation (CVM), the patient had reached stage CS4, indicating the completion of active craniofacial growth. However, because the correlation between CVM stage CS4 and midpalatal suture maturation is not clearly defined, a CBCT scan was performed to directly evaluate suture morphology. This approach adhered to the SEDENTEXCT guidelines, which recommend the justified and cautious use of CBCT imaging in orthodontics to ensure diagnostic accuracy while minimizing patient radiation exposure [8,9]. The use of two CBCT examinations was justified in accordance with SEDENTEXCT guidelines, as a pre-treatment CBCT scan was required for the diagnostic evaluation of midpalatal suture maturation as well as the post-treatment volumetric imaging was essential to discriminate between skeletal, dental, and periodontal effects in the case patient.

The maturation of the midpalatal suture was assessed following the protocol by Angelieri et al. [10] This method involves a detailed visual analysis of all axial CBCT slices of the palate to assign a maturation stage. Five stages (A to E) describe progressive suture fusion. The patient’s suture was classified as stage C, characterized by two parallel, scalloped, high-density lines in proximity, separated by small low-density spaces within the maxillary and palatine bones, showing an irregular pattern (Figure 3).

In addition to the midpalatal suture, the zygomaticomaxillary buttress was also qualitatively evaluated, as this site represents a major area of skeletal resistance to maxillary expansion. The buttress appeared continuous and well defined at baseline, consistent with moderate skeletal resistance. This partial assessment is in line with the diagnostic framework proposed by Watted et al. [11], although a complete multilevel assessment of all resistance sites was not possible because of the limited FOV dimensions.

### 2.7. Treatment Objectives and Treatment Plan

A comprehensive problem list for this case would therefore include: (i) transversal discrepancy with bilateral posterior crossbite, (ii) sagittal imbalance with mild Class III tendency, (iii) vertical hypodivergent pattern, (iv) anterior crowding, and (v) impaired sleep quality. Corresponding treatment objectives were to correct the transverse discrepancy through maxillary expansion, to improve occlusal relationships and arch coordination, to maintain periodontal health, and to achieve secondary improvements in airway function. The retention strategy prior to the fixed multibracket therapy was to maintain in situ the RPE for 8 months; the retention period allowed bone remodeling and the stabilization of the transversal width.

#### Treatment Alternatives

The patient declined all surgical options, including SARPE, MARPE, and any form of orthognathic surgery. Detailed information regarding these approaches was provided; however, the patient expressed concerns about surgical interventions and withheld consent. Given the patient’s favorable periodontal condition and a midpalatal suture classified as Stage C, a conventional tooth-borne RPE was chosen. The patient was thoroughly informed about the potential risks, benefits, and limitations of this approach, including the possibility of treatment failure. RPE was chosen over dentoalveolar expansion with a quad-helix appliance to reduce the risk of periodontal compromise. Additionally, the use of a jackscrew expander allowed for greater control during the activation phase.

### 2.8. Treatment Progress

An intraoral digital scan of both arches was obtained to fabricate the appliance. A Hyrax-type expander with four arms and bands on the upper first molars was constructed. Two anterior arms were contoured to adapt to the palatal surfaces of the upper premolars, while two posterior arms were soldered to the molar bands. The jackscrew used was a 13 mm screw from Leone S.p.A. (A2620—Sesto Fiorentino, Florence, Italy). After placement, the first and second premolars were bonded to the appliance using composite resin.

Activation of the appliance began the day after placement, following a two-phase protocol. In the first phase, the screw was turned four times daily—twice in the morning and twice in the evening—for one week, until overcorrection of the crossbite was achieved. In the second phase, activation was reduced to two turns per day, one in the morning and one in the evening, for an additional week, continuing until the palatal cusps of the maxillary molars contacted the lingual cusps of the mandibular molars. A total of 42 turns were performed during the entire activation period

Following expansion, the appliance remained in place for an 8-month retention period to allow for stabilization and bone remodeling.During the active expansion phase, the patient was monitored on a weekly basis to verify the presence of the midline diastema, assess soft-tissue tolerance, and record any occlusal change. Periodontal parameters, including gingival health and probing depth at the anchorage teeth, were evaluated at each visit to exclude inflammation or attachment loss. A follow-up CBCT scan was performed at the end of the active phase to confirm the separation of the midpalatal suture and to ensure parallelism of the expansion. No adverse effects such as mucosal irritation, discomfort, or mechanical loosening of the appliance were reported throughout the activation or retention period.

After retention, residual dentoalveolar discrepancies were addressed with a fixed multibracket appliance.

### 2.9. Measurements

#### 2.9.1. Qualitative Evaluation of Maxillary Expansion on Cone Beam Computed Tomography

The maxillary expansion was assessed through CBCT following the methodology described by Gurani et al. [12]. Imaging was performed before the activation of the expansion device (CB0) and after its removal (CB1), using the same radiographic unit for consistency. The CBCT scans were performed with a voxel size of 0.3 mm, 90 kVp, 10 mA, a 16 × 13 cm field of view, and an exposure time of 9 s. The effective dose per scan was approximately 50 µSv. DICOM data from both time points were imported into MIMICS 19 software (Materialise^®^, Leuven, Belgium), where a custom thresholding protocol was applied (lower threshold: −1024 HU; upper threshold: 566–3657 HU) to generate 3D volumetric reconstructions. (Figure 4) The resulting segmented STL files were then imported and superimposed in Geomagic Control (3D Systems, Rock Hill, SC, USA) using cranial base landmarks as reference points. Superimposition of CB1 onto CB0 was examined across axial, coronal, and sagittal planes. Special attention was given to the evaluation of transverse maxillary displacement, symmetry between hemiarches, and potential changes in nasal cavity width. The presence of buccal dental tipping and cortical bone alterations (e.g., fenestrations or dehiscences) were also investigated [13,14]. All measurements and visual assessments were independently reviewed and confirmed by two experienced clinicians (V.C. and D.G.).

To assess reproducibility, 20% of the measurements were repeated after a two-week interval, and intra- and inter-examiner reliability were calculated.

#### 2.9.2. Quantitative Two-Dimensional Analysis of Upper Palatal Expansion on CBCT

Sutural expansion measurements were performed at the palatal cusp tips of the first molars (M1). Total expansion (TE) was defined as the difference between post-treatment (T2) and pre-treatment (T1) measurements of intermolar width (IMW). As reported in Figure 5, intermolar maxillary width (IMW) was measured as the linear distance between the palatal cusp tips of the right and left first molars (M1) on a coronal cross-section passing through the center of M1. Similarly, palatal maxillary width (PMW) was defined as the linear distance between the palatal cortical surfaces of the right and left maxillary alveolar processes at the level of the first molars, measured on the same coronal section. Both IMW and PMW served as indicators of transverse dental and skeletal changes. In accordance with Walter et al., the MORE factor was calculated as the ratio between skeletal expansion at the midpalatal suture (SE) and intermolar width (IMW). SE was measured as the skeletal separation at the midpalatal suture, specifically at the M-point, defined as the midpoint between the anterior nasal spine (ANS) and posterior nasal spine (PNS) on an axial CBCT slice.

#### 2.9.3. Evaluation of Dental Effects: Alveolar Bending, Palatal Alveolar Angle, Dental Tipping and Bone Dehiscence

Two-dimensional (2D) measurements were performed to assess the dental and alveolar effects of maxillary expansion, specifically focusing on alveolar bone bending, dental tipping, buccal bone thickness, and the presence of alveolar defects. As shown in Figure 6, alveolar bone bending was quantified as the change in the PAA of the anchoring teeth—first premolars (P1), first molars (M1), or both—measured in degrees on coronal cross-sectional images at the midpoint of the teeth. An increase in the PAA was interpreted as buccally directed alveolar bending. Buccal tipping of the posterior teeth was evaluated by measuring the change in the DTA of molars and premolars, again expressed as the angular difference on corresponding coronal sections through the center of the crowns. Buccal bone thickness (BBT) was assessed at the level of P1, the mesiobuccal root of M1, and the distobuccal root of M1, based on the anchorage configuration. These measurements were taken from axial cross-sectional images passing through the furcation area of M1 and defined as the shortest perpendicular distance from the most buccal point of the root surface to the outer buccal cortical plate. Buccal bone dehiscences were examined following the protocol described by Baysal et al.: the full root length in serial cross-sections on both buccal and palatal aspects identify the presence or progression of alveolar defects [15]. Additionally, cortical bone integrity was visually evaluated to detect any buccal or palatal bone fenestrations, if the alveolar crest remained intact.

## 3. Results

The intraclass correlation coefficient (ICC) was >0.90 for all variables, indicating excellent agreement. Measurement error was consistently below 0.2 mm or 0.5°.

### 3.1. Orthodontic Results

After 14 days of treatment, a midline diastema occurred (Figure 7). The patient did not relate any pain symptoms or mucosal changes during non-surgical RME. The posterior crossbites were corrected, and the final molar relationship was acceptable. The midlines were coincident with each other and with the facial midline. The reperforming of PSQI highlighted an improvement of sleep quality from 7 to 3 points.

### 3.2. Qualitative Evaluation of Maxillary Expansion on Cone Beam Computed Tomography

As reported in Figure 8, the superimposition of CBCTs graphically represents the amount of expansion obtained. The superimposition analysis revealed several key features. In the coronal view, a lateral displacement of the upper first hemiarch was observed, indicating asymmetric expansion between the maxillary halves. No significant buccal tipping of the posterior teeth was detected, suggesting that the expansion was achieved primarily through skeletal rather than dental movement. Additionally, the width of the nasal cavity remained unchanged following treatment, indicating minimal transverse skeletal alteration at the nasal level: nasal width increases by 0.2 mm from 19.96 mm to 19.98 mm.

At the level of the zygomaticomaxillary buttress, a slight lateral displacement was observed while continuity was preserved, indicating partial transmission of expansion forces without structural disruption.

Cortical bone integrity appeared stable, with no appreciable differences observed in the buccal or palatal cortices before and after the expansion phase.

### 3.3. Quantitative Two-Dimensional Analysis of Upper Palatal Expansion on CBCT

Following treatment, CBCT analysis revealed significant transverse maxillary expansion. The PMW, measured between the palatal cortical plates at the level of the first molars, increased from 28.0 mm at T0 to 34.5 mm at T1, indicating a skeletal gain of 6.5 mm. Intermolar width (IMW), assessed as the linear distance between the palatal cusp tips of the right and left first molars, increased from 36.08 mm to 50.02 mm, resulting in a total expansion (TE) of 13.94 mm. The skeletal expansion (SE), measured at the M-point on an axial CBCT slice, was 7.53 mm. Based on this, the MORE factor, calculated as the ratio of SE to IMW, was 0.54 (54%), indicating that the majority of the transverse gain was attributable to skeletal expansion rather than dentoalveolar tipping.

### 3.4. Evaluation of Dental Effects: Palatal Alveolar Angle, Buccal Tipping Angle, Bone Dehiscences and Fenestrations

The dental tipping angle of the upper right first molar (DTA 16°) increased by 1.04°, from 89.1° to 91.6°. Similarly, the dental tipping angle of the upper left first molar (DTA 26°) increased by 2°, from 100.9° to 102.9°. The PAA of the upper right molar (PAA 16) increased by 2.6°, from 93.6° to 96.2°, whereas the PAA of the upper left molar (PAA 26) increased by 2.6°, from 109.6° to 112.2°.

In terms of buccal bone thickness, an increase of 0.1 mm was observed in the upper right first molar (M16), from 0.6 mm to 0.7 mm; an increase of 0.7 mm in the upper right second premolar (M15), from 1.6 mm to 2.3 mm; and an increase of 0.6 mm in the upper left second premolar (M25), from 0.7 mm to 1.3 mm. Conversely, a reduction of 0.3 mm was noted in the upper left first molar (M26), decreasing from 1.2 mm to 0.9 mm. Table 3 summarizes the measurements recorded at T0 and T1 for each evaluated site.

### 3.5. Clinical Evaluation of the Periodontal Status

As reported in Table 1, the periodontal evaluation at T1 confirmed the stability of almost all clinical indices, showing a reduced bleeding on probing and the improvement from 2 to 1 of the calculus index.

### 3.6. Follow-Up and Outcome Evaluation

All clinical and radiographic parameters confirmed the stability of the expansion achieved. CBCT measurements demonstrated a parallel opening of the midpalatal suture, with mean skeletal widening of approximately 5 mm at the alveolar crest and 4.8 mm at the nasal floor. Intermolar width remained stable within 0.3 mm of the post-expansion values, indicating minimal relapse. Periodontal probing depths and gingival health indices were unchanged compared with baseline, confirming tissue stability around the anchorage teeth. Functionally, the patient reported the disappearance of nocturnal snoring and improved nasal breathing during exercise. The PSQI score improved from 7 to 3, reflecting a clinically meaningful enhancement in sleep quality. No complications—such as soft-tissue irritation, appliance loosening, or root resorption—were observed during the active or retention phases. 

## 4. Discussion

### 4.1. Interpretation of the Results

Before interpreting the results, it is important to clarify case management in relation to the diagnostic records. Functional crossbite, skeletal asimmetry, and nasal obstruction were ruled out based on clinical examination and CBCT evaluation, ensuring that the transverse discrepancy was of skeletal origin. The patient’s midpalatal suture maturation was classified as Stage C, due to partial fusion of the midpalatal suture, thus justifying the indication for conventional tooth-borne rapid palatal expansion. The results of this case study demonstrate that tooth-borne RPE led to a huge skeletal effect, explained by the transverse gain at the level of the IMW. This increase was primarily attributed to the complete opening of the midpalatal suture, as confirmed by CBCT analysis and quantified using MORE factor. Although a slight increase in both the DTA and PAA was observed, MORE factor of 53% indicated that the expansion was predominantly skeletal rather than dentoalveolar. This is particularly noteworthy given the use of a tooth-borne expander in a skeletally mature patient—a context in which greater dental compensation might typically be expected. No signs of buccal alveolar bone dehiscence or fenestration were detected on post-treatment CBCT scans. These results suggest that the applied biomechanical forces were sufficient to achieve clinically meaningful skeletal effects without compromising the integrity of the buccal cortical bone.

In addition to midpalatal suture morphology, the zygomaticomaxillary buttress was also evaluated on CBCT slices as an auxiliary reference of skeletal resistance, in line with the multilevel diagnostic approach proposed by Watted et al. [11]. Although the imaging protocol was not optimized for a comprehensive assessment of all resistance areas, this additional evaluation supported the diagnosis of a favorable condition for conventional RPE in this patient.

The improvement of PSQI was associated with the increase in the palatal volume, while nasal cavity width remained unchanged, suggesting limited transverse skeletal change at the superior anatomical level [16]. Although significant skeletal expansion occurred at the palatal level, the nasal cavity width showed only minimal change (0.2 mm). This discrepancy may be explained by resistance at higher anatomical levels, as also observed by Baratieri et al. [17]. The reduction in PSQI score from 7 to 3 may be clinically relevant, as scores above 5 are typically associated with impaired sleep quality. However, this improvement should be interpreted with caution, as the PSQI represents a subjective self-reported measure, and no objective airway testing (such as rhinomanometry or polysomnography) was performed. The observed improvement may be associated with increased palatal vault volume and enhanced tongue posture, which could secondarily contribute to better airway patency, rather than nasal width changes alone. This hypothesis-generating result aligns with previous reports describing associations between maxillary expansion and improved sleep quality [18], but no causal relationship can be established from a single case. Furthermore, it should be noted that the assessment of skeletal resistance sites was partial, as the CBCT field of view (FOV) was limited and primarily focused on the midpalatal suture and zygomaticomaxillary buttress.

### 4.2. Generalization of Results in the Context of the Literature

The present case report illustrates the clinical indications for tooth-borne expansion in young adults who fall within the “gray zone” of skeletal maturation. Although the general consensus associates successful expansion with early or mid-pubertal stages, the quantitative CBCT findings presented here demonstrate that clinically relevant skeletal widening can still occur in carefully selected post-pubertal individuals. This observation reinforces the concept—outlined in previous works—that individualized diagnostic assessment of midpalatal suture morphology is more predictive than chronological age when determining the feasibility of non-surgical treatment.

These findings align with previous literature reporting successful skeletal expansion via conventional RPE in patients with favorable midpalatal suture morphology. As outlined by Angelieri et al., patients exhibiting stage C maturation can achieve predictable, non-surgical separation of the midpalatal suture [10]. Similarly, Baratieri et al. observed stable skeletal expansion outcomes in patients at early maturation stages, even without the use of skeletal anchorage [17]. From a periodontal perspective, the present results corroborate those reported by Gauthier et al. and Calil et al., who noted minimal periodontal compromise following both MARPE and conventional RPE [19,20]. In the current case, post-treatment CBCT imaging showed no evidence of dehiscence or fenestration across any anchorage teeth, despite some mild increases in angular measurements. The absence of visible cortical defects further supports the safety of conventional RPE when applied to carefully selected adult patients. While some studies have documented complications such as alveolar tipping or buccal bone loss, these effects were not observed in this case—likely due to favorable sutural conditions, precise activation protocols, and the absence of excessive force transmission to the buccal cortex [21,22]. Notably, recent systematic reviews have highlighted the positive effects of RPE on upper airway morphology, suggesting that this orthopedic intervention may contribute to improved respiratory function [23,24].

From an orthodontic perspective, skeletal stability remains the primary concern, since adults often present with reduced adaptive capacity of the midpalatal suture. From a maxillofacial surgical standpoint, SARPE still represents a more predictable option in cases with advanced suture maturation, albeit with higher morbidity and costs. From the sleep medicine perspective, several studies support the role of maxillary expansion in improving upper airway patency and reducing sleep-disordered breathing, which further strengthens the potential systemic benefits of this approach [25,26]. When compared with MARPE and SARPE, conventional tooth-borne RPE offers some advantages, including lower cost, reduced invasiveness, and greater patient acceptance, particularly in individuals who decline surgical procedures. However, in cases of advanced suture maturation, MARPE and SARPE provide more predictable skeletal expansion and long-term stability, albeit at the expense of higher morbidity and financial burden [27,28,29].

### 4.3. Strengths and Limitations

The principal limitation of this study lies in its single-subject design, which restricts the generalizability of the results and precludes the development of standardized clinical protocols. The absence of a control group and limited follow-up duration further constrain the ability to draw definitive conclusions regarding long-term stability and periodontal impact. Another important limitation is the short follow-up period. Long-term studies (≥5 years) have documented partial relapse of transverse expansion and variable periodontal changes following RPE, MARPE, and SARPE [25,26]. Therefore, the long-term stability of the present results cannot be assured. In addition, the present case is subject to other limitations, including the single-case design and potential selection bias, which restrict external validity.

The lack of objective airway function assessments (e.g., rhinomanometry, acoustic rhinometry, or volumetric CBCT analysis) limits interpretation of the sleep-related outcomes. As such, the improvement in PSQI should be considered an associated finding rather than a direct causal effect of expansion.

Although reproducibility was high, as confirmed by excellent ICC values (>0.90) and low measurement error (<0.2 mm/0.5°), the use of CBCT remains a limitation due to radiation exposure. Nevertheless, the effective dose (≈50 µSv per scan) was justified by the diagnostic need to assess skeletal, dental, and periodontal changes in this young adult patient. The absence of low-dose CBCT protocols or alternative volumetric airway assessments further limits the generalizability of the findings.

Finally, the short follow-up period prevents definitive conclusions on long-term skeletal stability and periodontal safety. These factors should be considered in light of reports describing variable skeletal effects and a higher incidence of periodontal risks when tooth-borne expanders are used in adults [19,20]. Nonetheless, the study presents several strengths. The use of CBCT for both pre- and post-treatment assessments, conducted with the same machine and operator, enhances measurement reliability. Additionally, the inclusion of detailed periodontal evaluation adds clinical relevance by addressing both therapeutic efficacy and biological safety. The integration of skeletal, dental, and periodontal metrics allows for a more comprehensive understanding of the treatment’s multidimensional effects.

### 4.4. Clinical Implications

This case highlights the importance of individualized treatment planning for young adult patients with transverse maxillary deficiency. From a broader clinical perspective, the present findings showed that, when midpalatal suture morphology corresponds to Stage C, a tooth-borne expander can still produce a predominantly skeletal response, ensuring periodontal safety and functional improvement. These outcomes reinforce the need for individualized assessment of skeletal maturation rather than rigid age-based criteria. In the so-called “gray zone”—patients with incomplete midpalatal suture maturation (stage C)—tooth-borne RPE may still represent a viable and conservative alternative to MARPE or SARPE, potentially reducing both invasiveness and treatment cost [10,19,20]. Accordingly, pre-treatment CBCT assessment of skeletal maturation remains essential to guide the therapeutic protocol [20]. 

### 4.5. Implications for Future Research

In the present case, the diagnostic process focused primarily on midpalatal suture morphology and the zygomaticomaxillary buttress for the assessment of skeletal resistance, in line with the multilevel diagnostic approach suggested by Watted et al. [11]. However, other anatomical structures involved in transverse resistance, including the pterygomaxillary junction and the nasal aperture pillars, were not systematically analyzed because the CBCT protocol employed was optimized for dentoalveolar and periodontal evaluation and did not include extended fields of view or standardized landmarks for these sites. For this reason, patient selection was mainly based on midpalatal suture maturation, periodontal stability, and clinical/occlusal findings. Future protocols should explore the limits of skeletal adaptability in late adolescents and young adults through a systematic appraisal of other resistance structures, including the zygomaticomaxillary buttress, pterygomaxillary junction, and nasal aperture pillars to further refine diagnostic accuracy and treatment planning [11].

Further investigations involving dedicated CBCT protocols, larger cohorts, and longer follow-up periods are necessary to compare treatment outcomes of RPE, MARPE, and SARPE across different stages of midpalatal suture maturation to highlight precise indications for each technique. 

## 5. Conclusions

### 5.1. Patient Perspective

The patient expressed satisfaction with the overall treatment outcome, particularly noting improved nasal breathing and the disappearance of nocturnal snoring. He reported minimal discomfort during appliance activation and valued the non-surgical nature of the approach. From his perspective, the therapy provided meaningful functional and quality-of-life benefits with a manageable treatment experience.

### 5.2. Hypothesis

In the present case, significant skeletal expansion was achieved without detectable adverse effects on the periodontal structures or anchorage teeth. CBCT evaluation confirmed minimal dental tipping, preserved cortical bone integrity, and stable nasal width, suggesting that expansion was primarily skeletal and well-tolerated. Nevertheless, further longitudinal and controlled studies are required to confirm long-term stability, periodontal safety, and functional outcomes. Moreover, the present findings were limited to a single case with short follow-up and should therefore be viewed as hypothesis-generating rather than conclusive.

### 5.3. Take-Home Messages

The present case showed that the tooth-borne RPE can induce a predominantly skeletal effect even in a young adult patient.The assessment of midpalatal suture morphology and maturation through CBCT-scan should be integrated with a clinical screening of the periodontal status to guide case selection in the post-pubertal “gray zone.”A conservative, non-surgical expansion in adults who decline skeletal anchorage or surgery can provide a clinically meaningful improvement in sleep quality (PSQI 7 → 3) and stable periodontal health.

## Figures and Tables

**Figure 1 healthcare-13-02854-f001:**
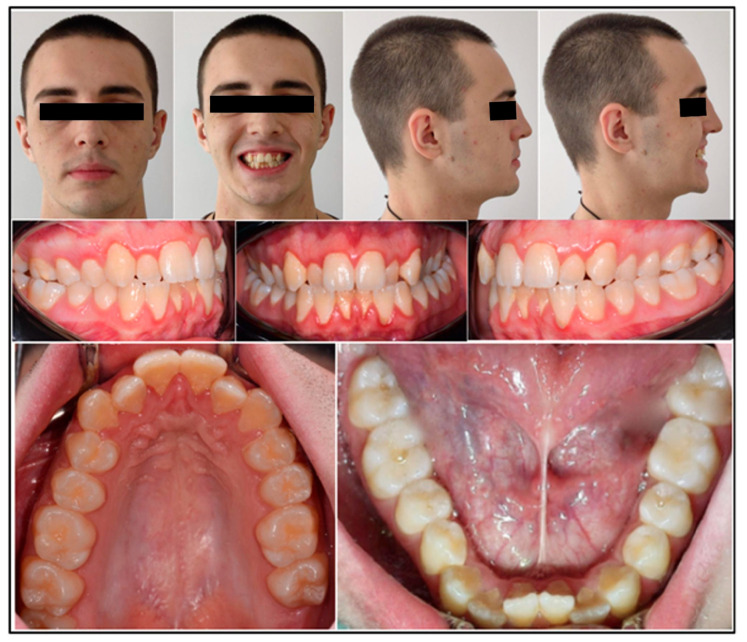
Pre-treatment intraoral and extraoral photographs.

**Figure 2 healthcare-13-02854-f002:**
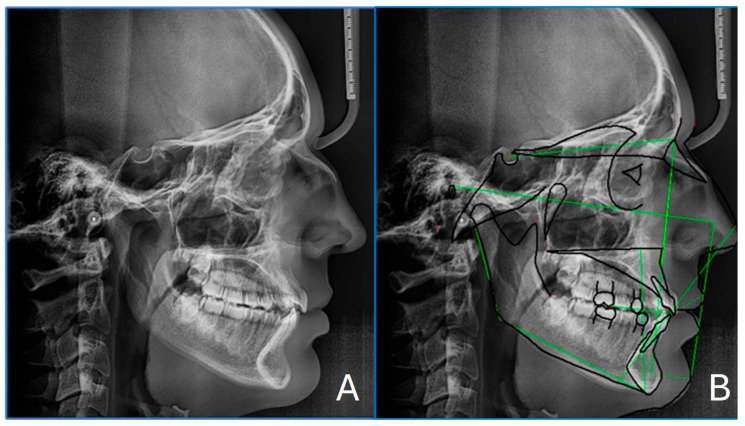
Pre-treatment lateral cephalogram (**A**) with cephalometric tracing in green(**B**).

**Figure 3 healthcare-13-02854-f003:**
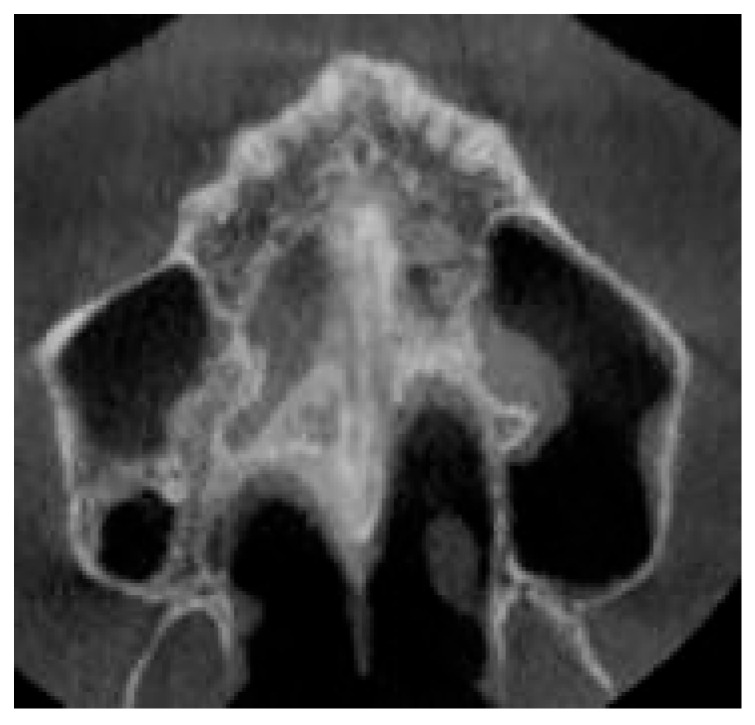
Evaluation of suture maturation before the palatal expansion. CBCT revealed stage C midpalatal suture maturation.

**Figure 4 healthcare-13-02854-f004:**
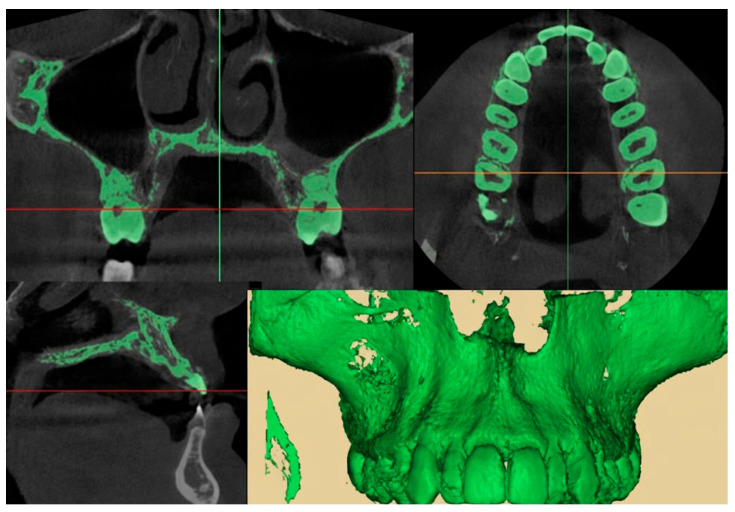
Segmentation protocol of the CBCT scan.

**Figure 5 healthcare-13-02854-f005:**
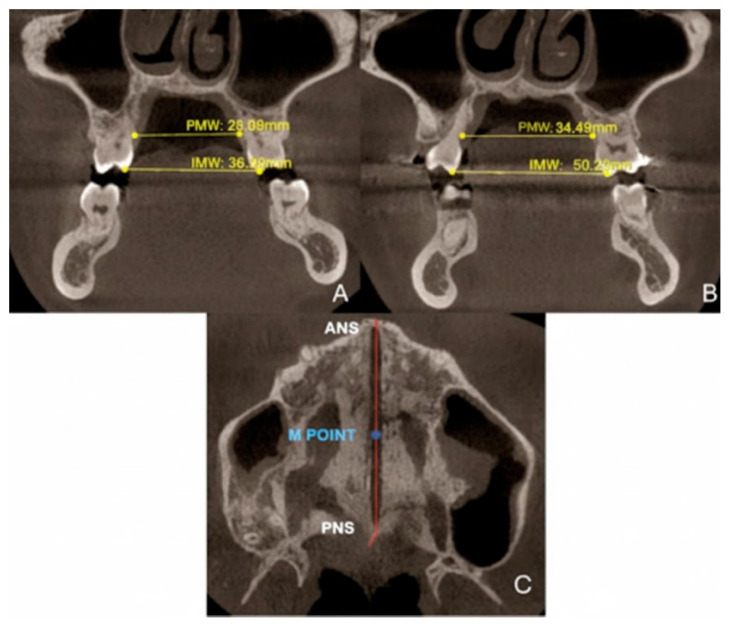
Variations in PMW and intermolar width IMW before (**A**) and after dental expansion (**B**). Reference points to evaluate the skeletal expansion (**C**).

**Figure 6 healthcare-13-02854-f006:**
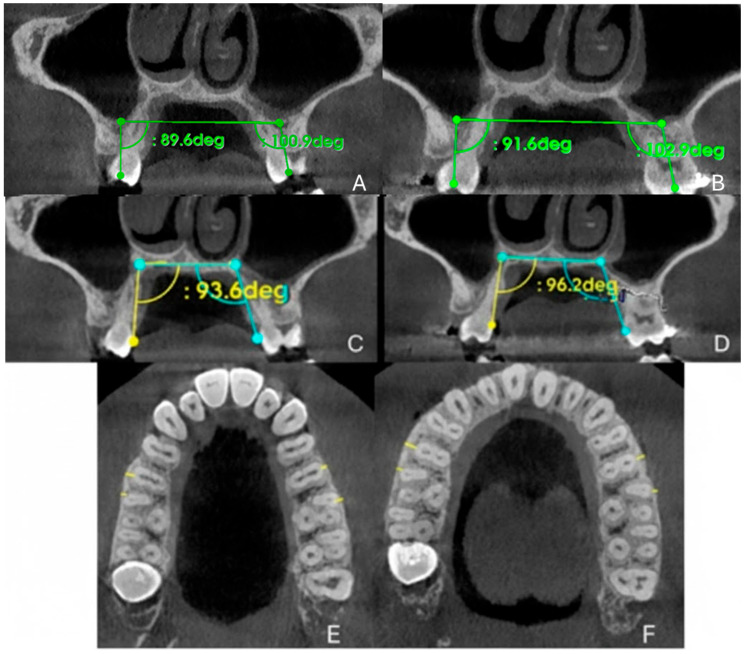
Tracing protocol of DTA, PAA and BBT before (**A**,**C**,**E**) and after (**B**,**D**,**F**) orthopedic expansion, respectively.

**Figure 7 healthcare-13-02854-f007:**
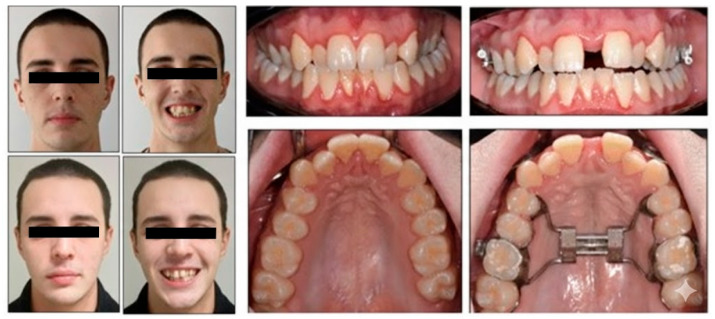
Comparison of extraoral and intraoral photographs before and after the activations of RME.

**Figure 8 healthcare-13-02854-f008:**
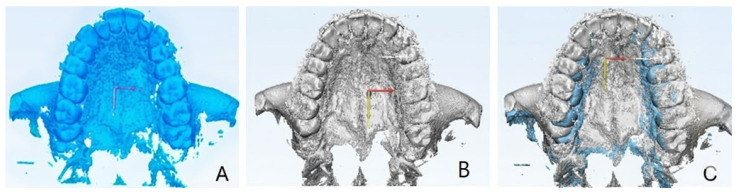
Segmentation of the maxillary bone. In blue: before RPE (**A**), in gray: after RPE (**B**). Superimposition (**C**) of the maxillary bone before and after RPE.

**Table 1 healthcare-13-02854-t001:** Baseline pre-treatment cephalometric measurements.

Parameter	Patient Value	Normal Range	Interpretation
SNA (°)	80.0	82 ± 2	Normal maxillary position
SNB (°)	80.2	80 ± 2	Normal mandibular position
ANB (°)	−0.1	2 ± 2	Skeletal class I
Wits appraisal (mm)	−0.1	0 ± 1	Skeletal class I
FMA (°)	20.1	25 ± 4	Hypodivergent growth pattern
Interincisal angle (°)	120.4	128 ± 5	Proclined incisors
A to N–Perp (mm)	−3.5	1 ± 2	Retruded maxilla (McNamara)
Pog to N–Perp (mm)	−3.1	0 ± 2	Retruded chin point (McNamara)
FH to AB (°)	86.5	81 ± 3	Skeletal Class III (McNamara)

**Table 2 healthcare-13-02854-t002:** Evaluation of periodontal status before and after the orthopedic treatment.

	Scoring System/Teeth and Surfaces	T0	T1
Modified gingival index	0: healthy 1: mild inflammation (partial unit) 2: mild inflammation (entire unit) 3: moderate inflammation 4: severe inflammation	2	2
Calculus index	0: no calculus 1: calculus covering up to 1/3 of the tooth surface 2: calculus covering up to 2/3 of the tooth surface and/or separate flecks of subgingival calculus 3: calculus covering more than 2/3 of the tooth surface and/or a continuous band of subgingival calculus	1/3	0
Plaque index	0: no plaque 1: separate flecks of plaque at the cervical margin of the tooth 2: thin continuous band of plaque (up to 1 mm) at the cervical margin of the tooth 3: band of plaque wider than 1 mm covering less than 1/3 of the crown of the tooth 4: plaque covering at least 1/3 but less than 2/3 of the crown of the tooth 5: plaque covering 2/3 or more of the crown of the tooth	1	1
Clinical attachment level	Distance from the cemento-enamel junction (CEJ) to the base of the pocket.	1	1
Bleeding on probing	resent/absent	P	A

Legend: T0 = pre-treatment score; T1 = post-treatment score; P = present; A = absent.

**Table 3 healthcare-13-02854-t003:** Measurements performed on CBCT.

Measurements	Pre-Treatment	Post-Treatment
PMW (mm)	28.04	34.5
IMW (mm)	36.08	50.02
SE (mm)	0.32	7.82
M16 (mm)	0.6	0.7
M15 (mm)	1.6	2.3
M25 (mm)	0.7	1.3
M26 (mm)	1.2	0.9
PAA 16 (°)	93.6	96.2
PAA 26 (°)	109.6	112.2
DTA 16 (°)	89.06	91.6
DTA 26 (°)	100.9	102.9

Legend: Pre-treatment = before orthopedic expansion, Post-treatment = At the end of orthopedic expansion; PMW = Palatal maxillary width, IMW = intermolar width, SE = Sutural Expansion, M16 = Difference in buccal bone thickness at the level of tooth 1.6, M15 = Difference in buccal bone thickness at the level of tooth 1.5. M25 = Difference in buccal bone thickness at the level of tooth 2.5, M26 = Difference in buccal bone thickness at the level of tooth 2.6, PAA DX = Palatal alveolar angle referred to tooth 1.6, PAA SX = palatal alveolar angle referred to tooth 2.6, DTA 16 = dental tipping angle of tooth 1.6, DTA 26 = dental tipping angle of tooth 2.6.

## Data Availability

Data and materials supporting the results or analyses presented in the present paper are available upon reasonable request to the corresponding author.

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
