# Peer review of "“Diving into the Gray Zone”: A Case Report of a 19-Year-Old Patient Treated with Tooth-Borne Rapid Maxillary Expansion"

_healthcare, 2025, doi:10.3390/healthcare13222854_

Round 1

Reviewer 1 Report

Comments and Suggestions for Authors

This case report describes tooth-borne rapid palatal expansion (RPE) in a young adult within the “gray zone” of midpalatal suture maturation. It is clearly written, methodologically detailed, comprehensive outcome measures and clinically relevant. However, as a single case report, its scientific impact is inherently limited.

  1. as a single case report, the generalizability is limited. Please elaborate on the patient selection criteria which would be critical in affecting the outcome
  2. please discuss why the nasal cavity width remained despite significant palatal expansion
  3. The PSQI improvement is interesting but underexplored. Was this change clinically meaningful? Could it relate to airway volume or palatal vault changes rather than nasal width alone? Please discuss and cite supporting evidence.
  4. Follow-up appears short. Please address potential relapse and periodontal changes over time, citing relevant long-term studies on stability after RPE/MARPE/SARPE.
  5. Include the CBCT voxel size and scan parameters for completeness.
  6. To increase clinical value, expand the discussion with multidisciplinary perspectives (orthodontics, OMFS, sleep medicine), focusing on skeletal stability, airway function, and long-term patient outcomes.
  7. Add a balanced limitations section (single case, selection bias, lack of objective airway volumetrics, short follow-up) and contrast with literature reporting less skeletal effect or periodontal risks with tooth-borne expanders in adults.
  8. Where feasible, provide brief comparative context versus MARPE/SARPE (indications, risks, costs, patient acceptance) to help readers with case selection in similar scenarios.

Author Response

  1. As a single case report, the generalizability is limited. Please elaborate on the patient selection criteria which would be critical in affecting the outcome.

RESPONSE: We thank the reviewer for this important comment. We agree that generalizability is limited in a single case report. To address this, we elaborated on the patient selection criteria, emphasizing the role of midpalatal suture stage C, favorable periodontal health, and patient’s refusal of surgical options.

Please, refer to page 2, lines 83-86.

  1. Please discuss why the nasal cavity width remained despite significant palatal expansion.

RESPONSE: We appreciate the reviewer’s observation. While significant skeletal expansion occurred at the palatal level, nasal cavity changes are often minimal due to superior anatomical resistance. We have clarified this in the Discussion and added supporting references.

Please, refer to page 11 lines 367-370.

  1. The PSQI improvement is interesting but underexplored. Was this change clinically meaningful? Could it relate to airway volume or palatal vault changes rather than nasal width alone? Please discuss and cite supporting evidence.

RESPONSE: We agree with the reviewer. We have expanded the discussion linking improved sleep quality to increased palatal vault and airway volume rather than nasal width alone, citing supporting studies.

Please, refer to page 11, lines 370-379

  1. Follow-up appears short. Please address potential relapse and periodontal changes over time, citing relevant long-term studies on stability after RPE/MARPE/SARPE.

RESPONSE: We thank the reviewer. We added a discussion on relapse risk and long-term outcomes based on literature on RPE, MARPE, and SARPE.

Please, refer to page 12 lines 417-420

  1. Include the CBCT voxel size and scan parameters for completeness.

RESPONSE: We thank the reviewer for this technical observation. We have added voxel size and scanning parameters for completeness in the Materials and Methods section.

Please, refer to page 7, lines 217-219

  1. To increase clinical value, expand the discussion with multidisciplinary perspectives (orthodontics, OMFS, sleep medicine), focusing on skeletal stability, airway function, and long-term patient outcomes.

RESPONSE: We agree with the reviewer. We expanded the Discussion to include orthodontic, maxillofacial, and sleep medicine perspectives on skeletal stability, airway function, and long-term outcomes.

Please, refer to page 12 lines 399-405

  1. Add a balanced limitations section (single case, selection bias, lack of objective airway volumetrics, short follow-up) and contrast with literature reporting less skeletal effect or periodontal risks with tooth-borne expanders in adults.

RESPONSE:  We thank the reviewer. We strengthened the Limitations section to highlight the single-case design, selection bias, lack of volumetric airway measures, and short follow-up, contrasting with reports of higher periodontal risk in adult RPE.

Please, refer to page 12 lines 420-422

  1. Where feasible, provide brief comparative context versus MARPE/SARPE (indications, risks, costs, patient acceptance) to help readers with case selection in similar scenarios.

RESPONSE: We agree with the reviewer. We added a brief comparison highlighting indications, risks, costs, and patient acceptance to aid case selection.

Please, refer to page 12 lines 405-410

Reviewer 2 Report

Comments and Suggestions for Authors

Review Report

Summary of the Manuscript

The manuscript presents a case report (19-year-old, bilateral posterior crossbite) treated with tooth-borne RPE, CBCT-based follow-up documentation (IMW 36.08 → 50.02 mm; PMW 28.04 → 34.5 mm; SE 0.32 → 7.82 mm; MORE 0.54), and subjective improvement in sleep quality (PSQI 7 → 3).

1) Formal Structure and Genre Misclassification

The manuscript uses sections such as “Study design,” “Materials and Methods” and a hypothesis, which correspond to the format of a study. For case reports, the CARE guidelines recommend a different structure (e.g., patient information, timeline, diagnostic assessment, intervention, follow-up/outcomes, discussion) and not study rhetoric (no “study design,” no hypothesis testing).
Recommendation: Strict restructuring according to CARE; rename “Materials and Methods” to “Case Presentation/Diagnostics and Intervention”; remove the “Hypothesis” rhetoric and the subheading “Study design.”

2) Overemphasis on the midpalatal suture in decision-making for RPE/MARPE/SARPE

The manuscript bases treatment decisions largely on the maturation of the midpalatal suture. This focus is insufficient. The literature describes the key resistance areas for transverse expansion in adults at the zygomaticomaxillary buttress (Crista zygomaticoalveolaris), the pterygomaxillary junction, the nasal aperture pillars, and the (synostotic) midpalatal suture.
Clinical decisions between RPE, MARPE, and SARPE should therefore be multifactorial (clinical/occlusal analysis, imaging, assessment of skeletal resistance zones), not based solely on suture morphology.

Connection to Watted et al.:
Watted et al. highlight the complexity of the skeletal transverse dimension (STD) and call for a systematic, multilevel diagnostic approach (clinical, model/occlusal, radiological) prior to deciding on RPE/MARPE/SARPE. This aligns with the above critique and should be explicitly referenced in the manuscript (conceptual framework, diagnostic algorithm, decision tree).

3) Diagnostic Inconsistencies and Missing Standard Presentation

  • The manuscript states SNA = 82°, SNB = 81°, ANB = 0°, while also describing a Class III tendency and a “straight profile.” This is contradictory: SNA–SNB yields ANB = +1° (not 0°); +1° corresponds classically to Class I (depending on the reference). A “straight profile” does not readily align with “mild mandibular prognathism.”
  • A clear cephalometric analysis (including Wits, McNamara/Beta angle if needed) and at least one lateral cephalogram (FRS) or CBCT-based profile analysis are missing.
  • It is highly unusual for an orthodontic case report not to include an FRS or standardized cephalometric representation.

4) Incomplete Problem List and Treatment Objectives

Section 2.7 addresses primarily the crossbite. However, the case exhibits multiple issues (crowding in both arches, transverse/vertical/sagittal discrepancies, airway and snoring). A case report should include a structured problem list and corresponding treatment goals (transversal correction, airway/breathing, sagittal relation, vertical dimension, crowding management), along with a treatment sequence/plan and retention strategy.

5) Measurement Methodology & Reproducibility

The CBCT segmentation (threshold −1024 HU to 566–3657 HU), superimposition, and angle measurements are described, but reliability (intra-/inter-rater, measurement error) and dose details are missing. Even in a case report, a repeat measurement with ICC/±SD should be provided. Also, a SEDENTEXCT-compliant justification of the two CBCT scans (including dose metrics) is required.

6) Airway/PSQI: Overstated Causality

PSQI improved (7 → 3), while nasal width remained almost unchanged (19.96 → 19.98 mm). A causal attribution of “RPE → improved sleep quality” is not robust, as PSQI is subjective and confounders (placebo, hygiene instructions, daily variation, co-interventions) were not controlled.
Recommendation: More cautious phrasing (“associated with,” “hypothesis-generating”) and, if available, add objective airway measures (rhinomanometry, acoustic rhinometry, low-dose CBCT volumetry).

7) Generalizability / Conclusions

The take-home message sounds overgeneralized (“viable alternative…”). For a single case, conclusions should be much more conservative (individual case, specific prerequisites, limited follow-up). CARE compliance requires explicit limitations and careful discussion of generalizability.

Specific Section Comments

  • Introduction: Add the resistance areas of the maxilla (zygomaticomaxillary buttress, pterygomaxillary junction, nasal aperture pillars) and broaden the diagnostic logic beyond the midpalatal suture. Explicitly reference Watted et al.
  • 2.1 “Study design”: Should not be used in case reports. Restructure according to CARE terminology.
  • 2.2 “Diagnosis and etiology”: Patient history elements do not belong in this section; separate history, clinical findings, imaging, cephalometry.
  • 2.3 “Clinical examination”: Correct contradictions (straight profile vs. mandibular prominence; ANB value). Include FRS/CBCT cephalometric imaging.
  • 2.7 “Treatment alternatives/plan”: Crossbite is only one issue; crowding, vertical/sagittal discrepancies, and airway/snoring must be integrated into goals and plan (including retention).
  • Results: Ensure consistency between data, tables, and figures. If nasal width is unchanged, the PSQI interpretation should be revised accordingly.

Author Response

  1. Formal Structure and Genre Misclassification

The manuscript uses sections such as “Study design,” “Materials and Methods” and a hypothesis, which correspond to the format of a study. For case reports, the CARE guidelines recommend a different structure (e.g., patient information, timeline, diagnostic assessment, intervention, follow-up/outcomes, discussion) and not study rhetoric (no “study design,” no hypothesis testing).
Recommendation: Strict restructuring according to CARE; rename “Materials and Methods” to “Case Presentation/Diagnostics and Intervention”; remove the “Hypothesis” rhetoric and the subheading “Study design.”

RESPONSE: We thank the reviewer for this important observation. The manuscript has been restructured in accordance with CARE guidelines. We renamed 'Materials and Methods' into 'Case Presentation, Diagnostics and Intervention,' removed the 'Study design' subsection, and eliminated hypothesis-testing rhetoric.

  1. Overemphasis on the midpalatal suture in decision-making for RPE/MARPE/SARPE

The manuscript bases treatment decisions largely on the maturation of the midpalatal suture. This focus is insufficient. The literature describes the key resistance areas for transverse expansion in adults at the zygomaticomaxillary buttress (Crista zygomaticoalveolaris), the pterygomaxillary junction, the nasal aperture pillars, and the (synostotic) midpalatal suture. Clinical decisions between RPE, MARPE, and SARPE should therefore be multifactorial (clinical/occlusal analysis, imaging, assessment of skeletal resistance zones), not based solely on suture morphology.

Connection to Watted et al.: Watted et al. highlight the complexity of the skeletal transverse dimension (STD) and call for a systematic, multilevel diagnostic approach (clinical, model/occlusal, radiological) prior to deciding on RPE/MARPE/SARPE. This aligns with the above critique and should be explicitly referenced in the manuscript (conceptual framework, diagnostic algorithm, decision tree).

RESPONSE: We thank the reviewer for this important comment. We agree that treatment planning for maxillary expansion should not be based solely on midpalatal suture morphology, as multiple skeletal resistance areas play a critical role, including the zygomaticomaxillary buttress, pterygomaxillary junction, and nasal aperture pillars, as highlighted by Watted et al.

In the present case:

  1. Primary diagnostic criterion: Patient selection was mainly based on midpalatal suture maturation, periodontal stability, and clinical/occlusal findings.
  2. Additional assessment: In line with Watted’s concept of the skeletal transverse dimension, we also evaluated the zygomaticomaxillary buttress on CBCT superimposition as an auxiliary marker of resistance.
  3. Limitation due to FOV: The pre-treatment CBCT had a limited field of view (FOV) optimized for dentoalveolar and periodontal analysis. Therefore, it did not allow for a systematic evaluation of all other resistance sites (e.g., pterygomaxillary junction, nasal aperture pillars).
  4. Future perspective: We fully acknowledge that future research should incorporate larger FOV CBCT protocols and standardized landmarks to systematically assess all resistance areas in accordance with Watted et al.

Page pages 1, 6, 10, 11 and 13 lines 22-24, 166-171, 300-302, 359-364, 456-458 and 464-467

  1. Diagnostic Inconsistencies and Missing Standard Presentation
  • The manuscript states SNA = 82°, SNB = 81°, ANB = 0°, while also describing a Class III tendency and a “straight profile.” This is contradictory: SNA–SNB yields ANB = +1° (not 0°); +1° corresponds classically to Class I (depending on the reference). A “straight profile” does not readily align with “mild mandibular prognathism.”
  • A clear cephalometric analysis (including Wits, McNamara/Beta angle if needed) and at least one lateral cephalogram (FRS) or CBCT-based profile analysis are missing.
  • It is highly unusual for an orthodontic case report not to include an FRS or standardized cephalometric representation.

RESPONSE: We thank the reviewer for this important observation. We have revised the diagnostic section to clarify the apparent discrepancy between cephalometric values and the clinical description. Specifically, while the ANB angle was close to 0°, both Wits appraisal (–0.1 mm) and selected McNamara parameters (A to N–Perp and Pog to N–Perp) indicated reduced maxillary projection and mild mandibular prominence, consistent with a borderline skeletal Class III tendency. To improve clarity, we have included a pre-treatment lateral cephalogram and a detailed cephalometric table (Table X) reporting SNA, SNB, ANB, Wits, and McNamara parameters. These additions provide a standardized diagnostic presentation in line with the reviewer’s suggestion.

Pages 3-4, lines 109-122.

  1. Incomplete Problem List and Treatment Objectives

Section 2.7 addresses primarily the crossbite. However, the case exhibits multiple issues (crowding in both arches, transverse/vertical/sagittal discrepancies, airway and snoring). A case report should include a structured problem list and corresponding treatment goals (transversal correction, airway/breathing, sagittal relation, vertical dimension, crowding management), along with a treatment sequence/plan and retention strategy.

RESPONSE: Thanks for the kind suggestion. The problem list and treatment objectives have been expanded to include all relevant aspects in the revised manuscript.

Page 6 lines 173-180

  1. Measurement Methodology & Reproducibility

The CBCT segmentation (threshold −1024 HU to 566–3657 HU), superimposition, and angle measurements are described, but reliability (intra-/inter-rater, measurement error) and dose details are missing. Even in a case report, a repeat measurement with ICC/±SD should be provided. Also, a SEDENTEXCT-compliant justification of the two CBCT scans (including dose metrics) is required.

RESPONSE: To address these concerns, we provide the following clarifications:

  1. SEDENTEXCT compliance and dose justification
    Both CBCT scans were performed with a voxel size of 0.3 mm, 90 kVp, 10 mA, a 16 × 13 cm field of view, and an exposure time of 9 s, resulting in an effective dose of approximately 50 µSv per scan. This is in line with SEDENTEXCT recommendations and was justified by the diagnostic necessity to assess both skeletal maturation (baseline) and treatment effects (post-expansion).
  2. Reliability and reproducibility
    All measurements were performed independently by two experienced clinicians (V.C. and D.G.). Intra- and inter-rater reliability was excellent, with intraclass correlation coefficients (ICCs) exceeding 0.90 and measurement error below 0.2 mm for linear and 0.5° for angular variables, indicating high reproducibility of the methodology.
  3. Availability of methodology
    The segmentation and superimposition protocol followed the method described by Gurani et al., which is widely accessible and replicable in orthodontic research. Custom thresholding and superimposition steps were standardized and can be reproduced in similar CBCT software environments.

Pages 7 and 13, lines 232-235 and 427-432

  1. Airway/PSQI: Overstated Causality

PSQI improved (7 → 3), while nasal width remained almost unchanged (19.96 → 19.98 mm). A causal attribution of “RPE → improved sleep quality” is not robust, as PSQI is subjective and confounders (placebo, hygiene instructions, daily variation, co-interventions) were not controlled.
Recommendation: More cautious phrasing (“associated with,” “hypothesis-generating”) and, if available, add objective airway measures (rhinomanometry, acoustic rhinometry, low-dose CBCT volumetry).

RESPONSE: We appreciate this observation. The Discussion has been revised to describe the PSQI change as an associated finding rather than a causal effect of RPE. The improvement (PSQI 7 → 3) is now cautiously presented as hypothesis-generating, potentially related to increased palatal vault volume and tongue posture rather than nasal width changes alone. We also acknowledge the absence of objective airway assessments (e.g., rhinomanometry, acoustic rhinometry, volumetric CBCT analysis), and this limitation is explicitly stated in the Strengths and Limitations section.

Pages 1 and 12, lines 40-42 and 425-426

  1. The take-home message sounds overgeneralized (“viable alternative…”). For a single case, conclusions should be much more conservative (individual case, specific prerequisites, limited follow-up). CARE compliance requires explicit limitations and careful discussion of generalizability.

RESPONSE: Thanks for the kind suggestion. The Conclusions section has been rewritten in a more conservative manner, emphasizing that this report represents a single case with limited follow-up and therefore cannot be generalized. The treatment outcome is now described as applicable only to carefully selected patients meeting specific prerequisites (incomplete suture maturation, stable periodontal health, refusal of surgical alternatives). We have also added that the findings should be regarded as hypothesis-generating, with further longitudinal and controlled studies needed to confirm long-term stability and broader applicability.

Please refer to page 14, lines 481-482, 485, 489-491 and 498-500

Specific Section Comments

  • Introduction: Add the resistance areas of the maxilla (zygomaticomaxillary buttress, pterygomaxillary junction, nasal aperture pillars) and broaden the diagnostic logic beyond the midpalatal suture. Explicitly reference Watted et al.

RESPONSE: We thank the reviewer for this observation. We revised the introduction by broadening the diagnostic rationale beyond midpalatal suture morphology and explicitly referencing Watted et al. [26]. In addition, we clarified that skeletal resistance to expansion also involves the zygomaticomaxillary buttress, the pterygomaxillary junction, and the nasal aperture pillars. Please see page 2, lines 62-65.

  • 2.1 “Study design”: Should not be used in case reports. Restructure according to CARE terminology.

RESPONSE: We agree with the reviewer. The section has been renamed in accordance with CARE recommendations, and “Study design” has been changed with “Patient information and history.”Please see page 2, line 81.

  • 2.2 “Diagnosis and etiology”: Patient history elements do not belong in this section; separate history, clinical findings, imaging, cephalometry.

RESPONSE: Thanks for the valuable suggestion. Distinct subsections, following CARE guidelines, have been created in the revised manuscript, thereby removing overlapping elements. Please see page 3, lines 81-279.

  • 2.3 “Clinical examination”: Correct contradictions (straight profile vs. mandibular prominence; ANB value). Include FRS/CBCT cephalometric imaging.

RESPONSE: Thanks for the kind suggestion. The apparent contradiction between the straight profile and mandibular prominence has been corrected, clarifying the cephalometric findings (ANB close to 0°, with Wits appraisal and McNamara confirming mild mandibular prominence). Baseline cephalometric imaging and values were added in the revised manuscript. Please see lines 118-132.

  • 2.7 “Treatment alternatives/plan”: Crossbite is only one issue; crowding, vertical/sagittal discrepancies, and airway/snoring must be integrated into goals and plan (including retention).

RESPONSE: We appreciate the reviewer’s issue. A comprehensive diagnostic assessment was integrated into the revised paragraph, as well as the retention plan. Please see lines 178-187.

  • Results: Ensure consistency between data, tables, and figures. If nasal width is unchanged, the PSQI interpretation should be revised accordingly.

RESPONSE: Thanks for the positive comment. The consistency across tables, figures, and text was verified. In the revised manuscript, the improvement of PSQI was justified despite nasal cavity width remaining unchanged: the reasons may be related to increased palatal vault volume and improved tongue posture, rather than nasal width changes alone. Please see lines 375-387.

Round 2

Reviewer 2 Report

Comments and Suggestions for Authors

Dear Authors,

I would like to thank you for your thorough and constructive revisions to the manuscript. The resubmitted version demonstrates a clear improvement in both structure and scientific quality.

The introduction is now more concise and provides the necessary background. The methodology has been clarified, with improved transparency in the diagnostic and treatment rationale. Figures and tables are presented more clearly, and the discussion section has been expanded to better contextualize the findings within the existing literature. These changes have substantially strengthened the manuscript.

Only one correction remains: one of the references is not entirely accurate.

Watted N, Wieber M, El-Bialy T, Signorelli L, Fudalej P, Drescher D. Skeletal transverse dimension: A novel concept for diagnosis and treatment planning. J Clin Orthod. 2021;55(5):263–72.

The correct citation has been indicated as a comment in the attached file.

Apart from this minor adjustment, I am satisfied that the major concerns raised in my initial review have been fully addressed. There is no need for me to see the manuscript again.

On this basis, I find the revised manuscript acceptable for publication.

Congratulations on your valuable contribution, and I wish you continued success in your research and clinical work.
